# Experimental Investigation of Mechanical Properties of Concrete Mix with Lightweight Expanded Polystyrene and Steel Fibers

**Syed Jahanzaib Shah** [1,2]**, Asad Naeem** [2,]*****, Farzad Hejazi** [3]**, Waqas Ahmed Mahar** [4,5,]***** and **Abdul Haseeb** [2]

1. Department of Civil Engineering, Universiti Putra Malaysia (UPM), Serdang 43400, Malaysia; gs56853@student.upm.edu.my
2. Department of Civil Engineering, Balochistan University of Information Technology, Engineering and Management Sciences (BUITEMS), Quetta 87300, Pakistan; haseebshady111@gmail.com
3. Faculty of Environment and Technology, The University of the West England, Bristol BS16 1QY, UK; farzad.hejazi@uwe.ac.uk
4. Department of Architecture, Faculty of Architecture and Town Planning (FATP), Aror University of Art, Architecture, Design and Heritage, Sukkur 65200, Pakistan
5. Sustainable Building Design (SBD) Lab, Department of Urban & Environmental Engineering (UEE), Faculty of Applied Sciences, Université de Liège, 4000 Liège, Belgium
* Correspondence: asad.naeem@buitms.edu.pk (A.N.); architectwaqas@hotmail.com (W.A.M.)

**Abstract:** The demand for lightweight aggregates in concrete compositions for diverse structural and non-structural applications in contemporary building construction has increased. This is to achieve a controllable low-density lightweight concrete, which reduces the overall structural weight. However, the challenge lies in achieving an appropriate strength in lightweight concrete while maintaining a lower unit weight. This research aims to evaluate the performance of lightweight concrete by integrating expanded polystyrene (EPS) as a partial replacement for coarse aggregate. Test specimens were cast by blending EPS with coarse aggregate at varying proportions of 0%, 15%, 30%, and 45%, while maintaining a constant water-to-binder ratio of 0.60. To enhance the bonding and structural capabilities of the proposed lightweight concrete mixes, reinforcement with 2% and 4% steel fibers by volume of the total concrete mix was incorporated. Silica fume was introduced into the mix to counteract the water hydrophobicity of EPS material and enhance the durability of lightweight concrete, added at a rate of 10% by weight of cement in all specimens. A total of 60 samples, including cylinders and beams, were prepared and cured over 28 days. The physical and mechanical properties of the lightweight EPS-based concrete were systematically examined through experimental testing and compared against a standard concrete mix. The analysis of the results indicates that EPS-based concrete exhibits a controllable low density. It also reveals that incorporating reinforcement materials, such as steel fibers, enhances the overall strength of lightweight concrete.

**Keywords:** expanded polystyrene; lightweight; steel fiber; silica fume; compressive strength; flexural strength

## 1. Introduction

Lightweight concrete (LWC) has gained more interest and is increasingly being explored by researchers due to its low density in contrast to conventional concrete. Interest in LWC has increased as an alternative solution to normal concrete due to its low dead load and self-weight of structural elements. As a result, smaller sections can be achieved. According to a study, the compressive strength and bulk density of low-strength but lightweight concrete ranges from 7 to 18 MPa and 800 to 1400 kg/m$^3$, respectively [1]. In past decades, researchers have tried to use different alternative materials as substitutes for aggregates to prepare LWC that can achieve acceptable strength with lower self-weight [2,3]. Initially, aggregates made from expanded fly ash, clays, preprocessed shales, and those that come from natural porous volcanic sources were used to lower the density of LWC [4].

In the beginning, lightweight concrete was mainly used as an insulating material with air-entrained mixes of volcanic ash and hydrated lime to lessen the overall weight of the material [5]. These low-density artificial lightweight aggregates have been used in concrete with varying degrees of success in density reduction. In recent years, expanded polystyrene (EPS) has been used as an alternative material to aggregates. The concrete industry has paid more attention to EPS due to its adequate low density, relative strength, and good thermal resistance. Expanded polystyrene is a lightweight cellular plastic with small spherical particles of 98% air and a wide range of densities (10 to 20 kg/m$^3$). The properties it possesses include having a closed-cell nature, being lighter weight and nonporous, and hydrophobicity [6]. The utilization of EPS aggregate in concrete is mainly to reduce its overall weight in comparison to conventional concrete, which has a large self-weight with a low strength-to-weight ratio and substandard performance of thermal insulation [7]. The keen innovation in the advancement of EPS-based concrete was the preparation under air-entrained conditions using lightweight aggregates (LWAs) of polymeric particles having bulk densities of about 16 to 160 kg/m$^3$. From the experimental work, it was observed that EPS beads of smaller sizes yielded concrete having a reasonable strength in the absence of additives [2]. In addition, 30% EPS incorporation by volume of self-compacted concrete could reduce the density and compressive strength up to 30% and 40%, respectively [8]. Similarly, in another study, the utilization of EPS in concrete mix was examined. It was reported that with the addition of EPS in concrete without a superplasticizer, the density decreased by about 11.3% and 16.2% vice versa [9]. It was also reported that 5% EPS substitution as fine aggregate in concrete resulted in 16% lower compressive strength in comparison to control specimen strength. However, by increasing the EPS content up to 10%, the tensile strength was enhanced by 43% [10]. Other research also demonstrated EPS's effectiveness in enhancing concrete's durability and mechanical properties [11]. So, EPS is a discarded waste product like other various types of waste, and it has superior physical and mechanical capabilities that can be incorporated in concrete mix at optimal levels without compromising the strength of concrete in order to meet the demands for modern buildings and construction [12–14].

In addition, like other cementing agent materials, LWC has also brittle characteristics [15]. This became clear when diagonal tension or shear failure was prominent in LWC. The brittleness of such lightweight concrete mix must be reduced with acceptable physical and mechanical properties [16]. To achieve a ductile nature in materials, numerous studies have demonstrated the use of discrete fibers as reinforcement materials in concrete with reasonable performance [17]. Similarly, when fibers of a hooked-end nature were incorporated in concrete with varying contents of 0.0% to 1.5%, it resulted in a split tensile strength of 10% to 18% higher [8]. At 2.0 vol% steel fiber, the compressive strength increased up to 20% [18]. Material fatigue strength also increases with the introduction of steel fibers by reducing crack opening [19]. Moreover, the steel fibers create a network structure in the concrete matrix, which effectively prevents segregation and cracks due to plastic shrinkage [20].

Furthermore, adding supplementary cementitious materials (SCMs) alters the inner structure of lightweight concrete, increasing the concrete mix's brittleness and resistance against cracking [21]. Recent research studies have proven that silica fume can effectively limit the inner concrete matrix by its bridging action. It reported increased strength development by adding about 10% condensed silica fume [11]. It was observed that the substitution of silica fume and pozzolans up to 5% and 15% by weight of cement, respectively, showed increases in the compressive strength. The minute particles of SCMs work as microstructural modifiers, reducing the void spaces in the cement matrix. Similarly, silica fume works as an effective pozzolan, chemically reacting to produce calcium silicate hydrate (C-S-H) and increasing the mechanical properties of concrete mix as a result.

This study explores lightweight EPS as a substitute for coarse aggregate in low-density, low-strength, but lightweight concrete. However, freshly mixed EPS-based concrete faces challenges like segregation due to its lightweight and hydrophobic nature, impacting its workability. Optimal aggregate replacement levels affect concrete strength, and excessive

substitution weakens it. Investigating EPS as a substitute is promising, but ongoing research mainly focuses on reducing density and imposing strength reduction challenges. Our study assesses the structural performance of lightweight concrete properties with varying EPS levels (0%, 15%, 30%, and 45%), steel fibers, and silica fume, aiming to address segregation and enhance bonding. The findings highlight that EPS-based concrete with steel fibers and silica fume shows potential with low density and higher flexural strength, and an optimal lightweight concrete design mix is proposed.

## 2. Materials and Methods

### 2.1. Materials

In this study, ordinary Portland cement (OPC) was used as a binder material according to ASTM C150 standards [22]. It was uniform gray in color and free from hard lumps. River sand was used as the fine aggregate, which conformed to the standard specifications for fine aggregate materials. After drying, it was passed through a sieve to remove any roots and debris. Locally available natural aggregate of 14 mm particle size was used in compliance with ASTM C33/33M-18 [23]. Sieve analysis was performed to confirm the standard particle size. Portable tap water was used for mixing the concrete ingredients and for curing purposes to fulfill the requirement of ASTM C1602/1602M [24]. Commercially available EPS beads were used. The lightweight EPS beads were small impermeable balls of 2–3 mm diameter with a bulk density of 20 kg/m$^3$. The physical properties of the mixed proportions are presented in Table 1.

**Table 1.** Physical properties of ordinary Portland cement and fine and coarse aggregates.

| Properties | Materials | | | |
|---|---|---|---|---|
| | OPC | EPS | Fine Aggregates | Coarse Aggregates |
| Specific gravity | 3.6 | - | 2.44 | 2.85 |
| Fineness (%) | 2.43 | - | 2.64 | 7.32 |
| Color | Grey | - | Dark | Dark |
| Soundness | 1.10 | - | - | - |
| Porosity (%) | - | - | 0.06 | 0.90 |
| Moisture content | - | - | 3.56 | 0.20 |
| Water Absorption (%) | - | 4% by vol. | 1.2 | 0.76 |
| Compacted bulk density (kg/m$^3$) | - | 20 | 1681.81 | 1803.69 |
| Loose bulk density (kg/m$^3$) | - | - | 1630 | 1571.69 |
| Standard consistency (%) | 27.3 | - | - | - |
| Initial setting time (min) | 120 | - | - | - |
| Final setting time (min) | 320 | - | - | - |
| Density (kg/m$^3$) | - | 13 | - | - |

To produce EPS-based concrete, the coarse aggregate was replaced with EPS at 0%, 15%, 30%, and 45% in the concrete mix. In addition, to reduce the brittleness of the concrete mixture, steel fibers (SFs) of 1 mm in diameter and 50.8 mm in length were added at 2% and 4% by weight of concrete mixed. The SFs were obtained from annealed wires via a process of thermal annealing and from burnt wires during the process of iron settings. The typical physical appearance of these materials is presented in Figure 1a–c. Finally, commercially available supplementary cementitious material (SCM), such as silica fume, was utilized to adjust the EPS-based concrete flow. The chemical compositions of OPC and SFs are summarized in Table 2.

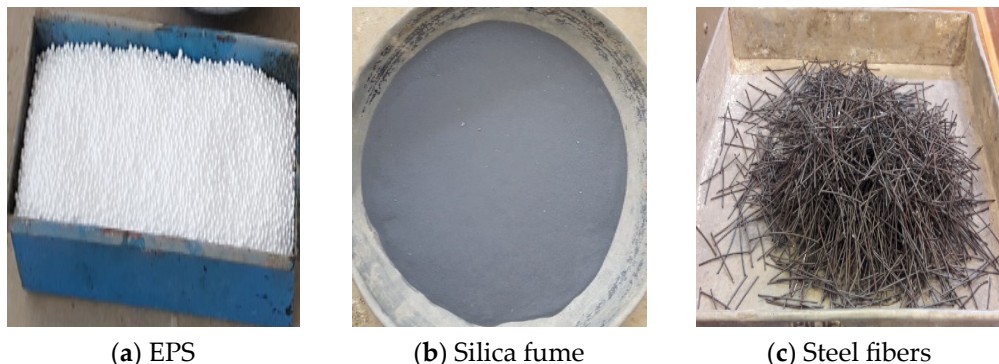

(**a**) EPS        (**b**) Silica fume        (**c**) Steel fibers

**Figure 1.** Appearance of materials.

**Table 2.** Chemical composition of OPC and silica fume.

| Chemical Composition | Weight (%) | |
|:---:|:---:|:---:|
| | **OPC** | **Silica Fume** |
| CaO | 62.3 | 1.2 |
| $SiO_2$ | 20.6 | 91 |
| $Al_2O_3$ | 5.6 | 1.3 |
| $Fe_2O_3$ | 3.4 | 1.6 |
| MgO | 3.6 | 1.3 |
| $SO_3$ | 2.4 | 0.1 |

## 2.2. Mix Proportions

In this study, concrete mix of M15 grade was used for the experimental design. Mix proportions of 1:2:4 were used to prepare all concrete specimens. Two main variables were adopted, namely EPS as a replacement for coarse aggregate ranging from 0%, 15%, 30%, and 45%, and steel fibers at proportions of 2% and 4% by weight of concrete mixed, as shown in Table 3. A water-to-binder ratio of 0.6 was used in the concrete mixes. Further, silica fume was incorporated at 10% by weight of cement in all concrete mixes. Prior to mixing, the molds were assembled and lubricated appropriately for the safe removal of hardened concrete samples. A total of 60 samples were prepared for each proportion, comprising 30 cylinders and 30 beam samples. The test samples were made in concrete cylinders ($\phi$150 $\times$ 300 mm$^3$) and concrete beams (150 $\times$ 150 $\times$ 450 mm$^3$).

**Table 3.** Details of specimen casting.

| EPS % | Cured Days | Steel Fiber % | No. of Cylinders | No. of Beams |
|:---:|:---:|:---:|:---:|:---:|
| 0% | 28 | 0 | 3 | 3 |
| 15% | 28 | 0 | 3 | 3 |
| | | 2 | 3 | 3 |
| | | 4 | 3 | 3 |
| 30% | 28 | 0 | 3 | 3 |
| | | 2 | 3 | 3 |
| | | 4 | 3 | 3 |
| 45% | 28 | 0 | 3 | 3 |
| | | 2 | 3 | 3 |
| | | 4 | 3 | 3 |

## 2.3. Mixing Process and Fabrication of Specimens

A standard mixing sequence for the making and curing of concrete specimens was adopted, which complied with ASTM C192 standard [25]. Initially, by using a laboratory

concrete mixer, all of the ingredients of concrete were blended, and then normal tap water was added to complete the mix. The mixer machine was used to obtain a homogenous mixture. Thereafter, the oiled molds for each specimen category were filled in three layers and compacted, as shown in Figure 2a,b. Concrete specimens of cylinders and beams were cast and kept at room temperature for 24 h, as shown in Figure 2c. All of the molds were sealed with a plastic sheet to avoid water loss due to evaporation. After a day, the concrete specimens were de-molded and placed in water for a curing period of 28 days. After all steps, three replicates for each sample were used for testing.

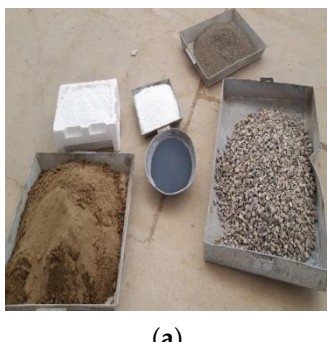 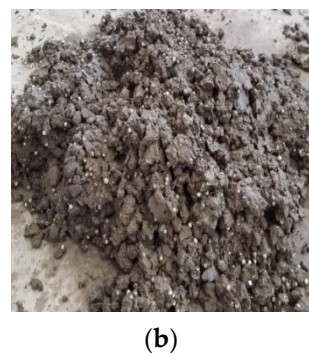 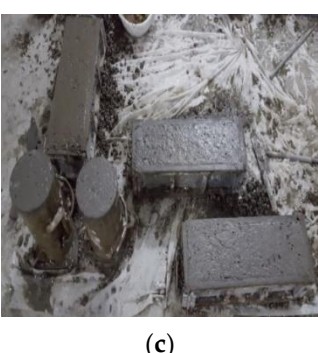

(**a**)                                        (**b**)                                        (**c**)

**Figure 2.** Preparation of test specimens. (**a**); Constituents of EPS-based concrete; (**b**) Mixing; (**c**) Casting of cylinders and beams.

*2.4. Test Methods*

2.4.1. Physical Properties of Materials

The distribution of aggregate particles by size was evaluated in accordance with the test procedure outlined in ASTM C136 [26]. Similarly, the tests for bulk density and porosity were accomplished using the standard provisions of ASTM C29 [27]. To check the properties of fresh concrete, consistency tests for all mix proportions were adopted by using a slump test conforming to ASTM C143 [28]. The workability of all EPS-based concrete mixes was determined and compared to the normal mix.

2.4.2. Compressive Strength Test

The compressive strength of concrete is the ability of concrete structural elements to carry the load without tending to crack or deflect. In this research, the cylinder specimens with dimensions ($\phi$150 $\times$ 300 mm$^3$) were used for the compressive strength test following ASTM C39 [29]. The tests were carried out on the specimen after the 28-day period of curing. A universal testing machine (UTM, Shimadzu Corporation, Kyoto, Japan) was used for this test. Three replicates were used for each mix. The average results of three specimens for each proportion were used for the comparative analysis. Prior to the compressive strength test, each cylinder's mass and volume were measured to determine its density.

2.4.3. Flexural Strength Test

The flexural strength test was conducted on beams having dimensions of 150 $\times$ 150 $\times$ 450 mm$^3$ at 28 days under center-point loading in accordance with ASTM C293 [30]. The test measures the resistance capacity of specimens against bending failure. The test indicates the modulus of the rapture of specimens. A total of 30 beam specimens with 9 samples for each proportion of EPS-based concrete were tested. For the results and analysis, the average value of three samples for each proportion was noted.

## 3. Results and Discussion

### 3.1. Physical Properties

3.1.1. Density

The saturated surface dry (SSD) densities of concrete specimens with varying proportions of EPS and steel fibers are shown in Figure 3. The figure illustrates that the density of specimens decreased with an increase in EPS content in the mix. The SSD density was reduced from about 1935 kg/m$^3$ to 925 kg/m$^3$ with an addition of 15% to 45% EPS, respectively, which could be classified as lightweight concrete. The experimental results showed the maximum reduction in SSD density by mixing a 45%/55% ratio of coarse aggregate/EPS, and the density of hardened concrete was reduced by 53% when compared to the standard concrete sample. For 15% and 30% EPS substitution to coarse aggregate, the reduction in density was about 20% and 35%, respectively. The density reduction was attributed to the weightless nature of EPS beads and the extremely hollow microstructure with lower specific gravity. The summary of all experimental results is shown in Table 4.

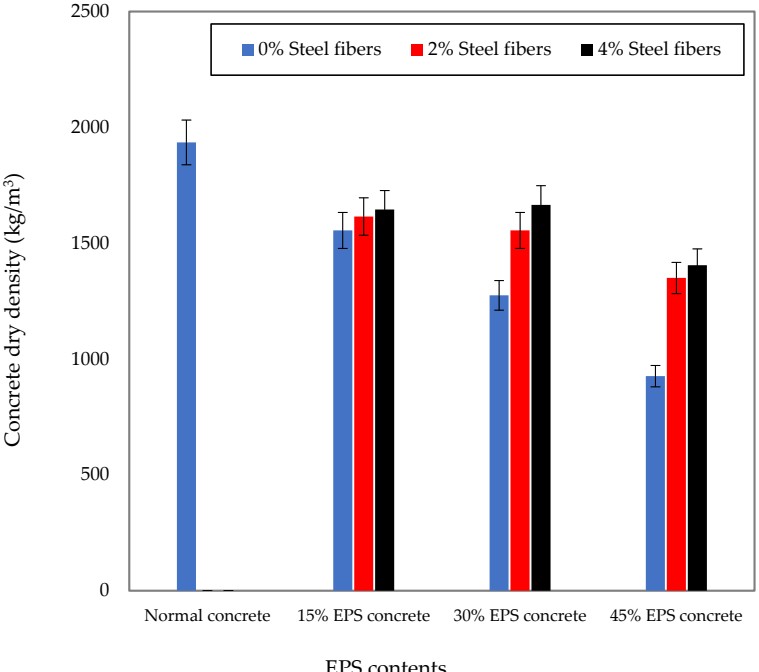

**Figure 3.** Effect of EPS proportions on the concrete density.

**Table 4.** Concrete density with different contents of EPS and steel fibers.

| Sample No. | Concrete with EPS (%) | Steel Fibers (%) | Density (kg/m$^3$) | | | |
|---|---|---|---|---|---|---|
| | | | Test 1 | Test 2 | Test 3 | Mean |
| 1 | 0% | 0% | 1940 | 1930 | 1935 | 1935 |
| 2 | 15% | 0% | 1560 | 1550 | 1555 | 1555 |
| 3 | 15% | 2% | 1610 | 1620 | 1615 | 1615 |
| 4 | 15% | 4% | 1650 | 1640 | 1645 | 1645 |
| 5 | 30% | 0% | 1270 | 1280 | 1275 | 1275 |
| 6 | 30% | 2% | 1550 | 1560 | 1555 | 1555 |
| 7 | 30% | 4% | 1570 | 1560 | 1665 | 1665 |
| 8 | 45% | 0% | 930 | 920 | 925 | 925 |
| 9 | 45% | 2% | 1340 | 1360 | 1350 | 1350 |
| 10 | 45% | 4% | 1410 | 1400 | 1405 | 1405 |

3.1.2. Workability

The EPS-based lightweight concrete's workability was evaluated in terms of slump test at a constant water: binder ratio of 0.60. The values of the slump measurements are

reported in Table 5. All proportions of mixes had slump values in the range of 4–40 mm (Figure 4). The concrete mix with 15% EPS content and without steel fibers had a slump value of 33 mm, which was 20% less than standard concrete. This could be due to the lightweight and hydrophobic nature of EPS, which tends to reduce the water absorption capacity of the concrete mixture as a result EPS particles floating up due to segregation and bleeding. Although at this value, the mixes were flexible and easily worked without compaction. Furthermore, the testing results indicated that the workability of concrete at the plastic stage was affected by the contents of EPS and steel fibers. It was observed that the slump of fresh concrete decreased with the addition of steel fibers because steel fibers held all of the ingredients together without segregation. The steel fibers contributed to decreasing the slump value of the concrete mix used for the specimens and helped to reduce the bleeding. When steel fibers were utilized as 4% of concrete with the same EPS dosage of 15%, there was about a 35% reduction in the slump value. Moreover, the addition of silica fume also reduced the slump value due to its pozzolanic properties. Silica fume reacts with calcium hydroxide to form additional cementitious compounds. This reaction can help control bleeding by absorbing excess water and preventing it from rising to the surface.

**Table 5.** Results of the slump cone test with proportions of EPS and steel fibers.

| Sample No. | Concrete with EPS (%) | Steel Fibers (%) | Water/Binder | Slump Value (mm) |
|---|---|---|---|---|
| 1 | 0% | 0% | 0.60 | 40 |
| 2 | 15% | 0% | 0.60 | 33 |
| 3 | 15% | 2% | 0.60 | 32 |
| 4 | 15% | 4% | 0.60 | 18 |
| 5 | 30% | 0% | 0.60 | 25 |
| 6 | 30% | 2% | 0.60 | 13.5 |
| 7 | 30% | 4% | 0.60 | 12.4 |
| 8 | 45% | 0% | 0.60 | 12 |
| 9 | 45% | 2% | 0.60 | 11 |
| 10 | 45% | 4% | 0.60 | 04 |

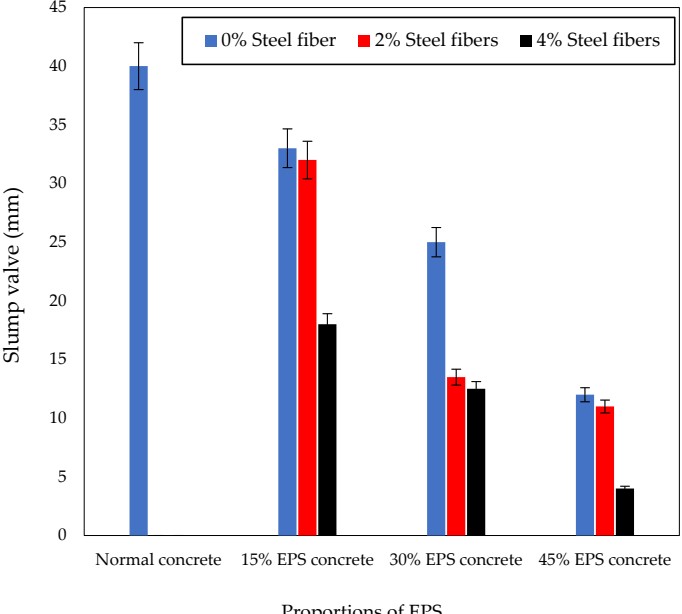

**Figure 4.** Workability of concrete with proportions of EPS and steel fibers.

*3.2. Mechanical Properties*

3.2.1. Compressive Strength

The 28-day compressive strengths of normal concrete with different contents of EPS and steel fibers are illustrated in Table 6. Figure 5 shows EPS-based concrete samples and their failure pattern. The incorporation of EPS solely resulted in reduced compressive strength within the test specimens. It decreased by 32%, 50%, and 71.1% in sample numbers 2, 5, and 8, respectively, compared to standard concrete, as illustrated in Figure 6.

**Table 6.** The compressive strength of concrete with various proportions of EPS and steel fibers at 28 days.

| % of EPS & Steel Fibers | | Compressive Strength (MPa) | | | |
|---|---|---|---|---|---|
| **EPS** | **SF** | **Test 1** | **Test 2** | **Test 3** | **Mean** |
| 0% | 0% | 15.648 | 18.271 | 17.800 | 17.240 |
| 15% | 0% | 12.414 | 12.482 | 12.241 | 12.379 |
| 15% | 2% | 15.637 | 15.846 | 15.526 | 15.67 |
| 15% | 4% | 14.353 | 14.185 | 14.240 | 14.259 |
| 30% | 0% | 7.588 | 7.243 | 7.347 | 7.393 |
| 30% | 2% | 8.308 | 8.650 | 8.540 | 8.499 |
| 30% | 4% | 8.262 | 8.927 | 8.098 | 8.429 |
| 45% | 0% | 5.186 | 5.518 | 5.077 | 5.260 |
| 45% | 2% | 6.158 | 5.891 | 6.217 | 6.089 |
| 45% | 4% | 4.756 | 4.577 | 4.555 | 4.629 |

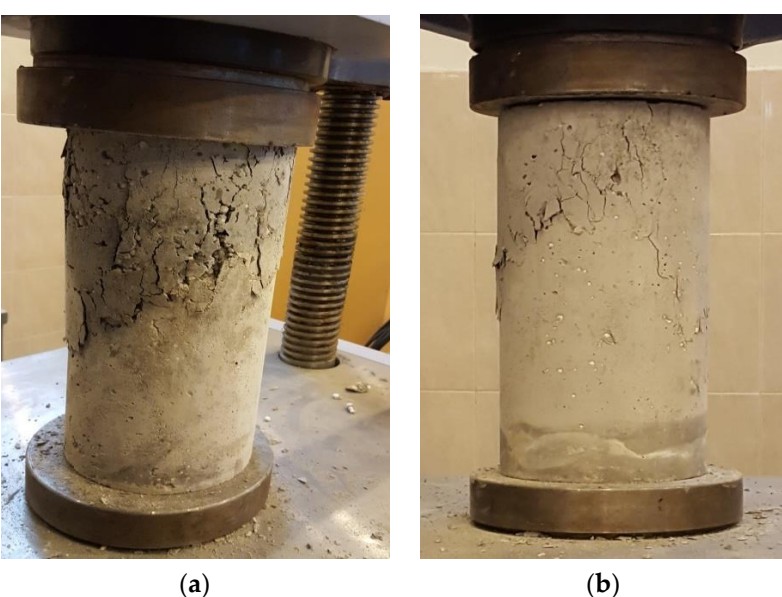

(**a**)  (**b**)

**Figure 5.** Compressive strength test. (**a**) EPS-based concrete sample failure pattern without steel fibers; (**b**) EPS-based concrete sample failure pattern with steel fibers.

The specimens made with the incorporation of a higher content of EPS had the least compressive strength. This reduction could be attributed to the lightweight and hydrophobic nature of EPS [31], which tended to reduce the water absorption capacity of the concrete mixture and float up, with segregation occurring as a result. The compensation for the reduction in concrete strength with the incorporation of EPS has also been discussed in a previous study. For example, EPS with 15% replacement of coarse aggregate reduced the compressive strength of lightweight concrete up to 45% [32]. Steel fibers were incorporated to amplify the compressive strength to counter the effect of improper bonding in EPS-based concrete. For instance, in concrete with 15% EPS and 2% steel fibers, the strength was slightly reduced to 10% compared with normal concrete. The concrete samples that

contained 30% EPS with 4% steel fibers had 34.6% less compressive strength than normal concrete. Similarly, the compressive strength of concrete that contained 45% EPS with 4% steel fibers showed 74.6% less strength than normal concrete.

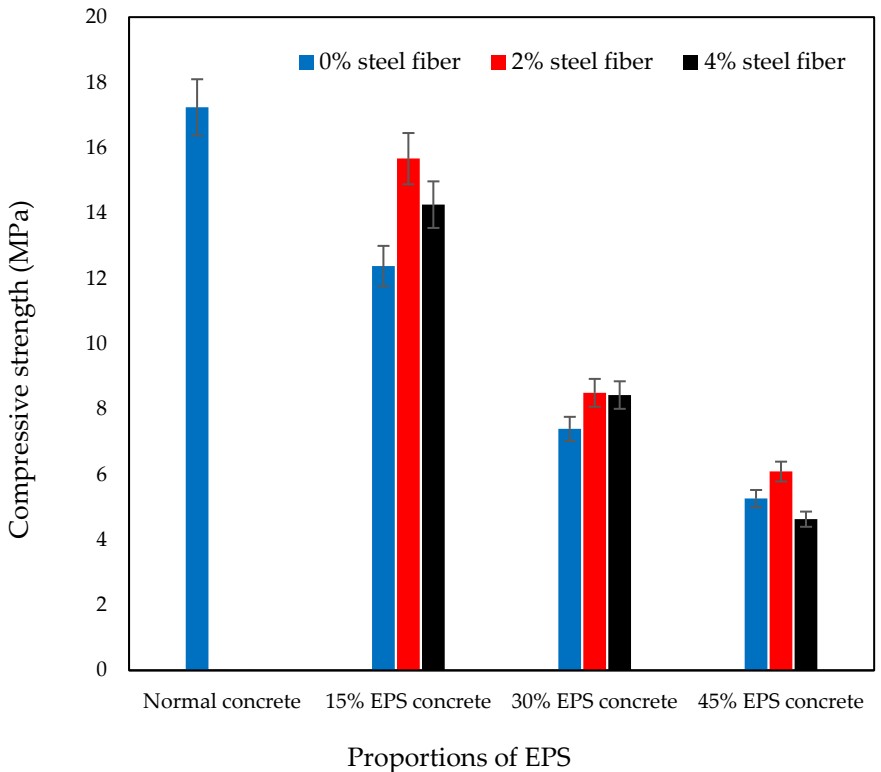

**Figure 6.** Compressive strength at 28 days of concrete with different contents of EPS and steel fibers.

Moreover, the reduction in the compressive strength of concrete increased as more content of EPS was substituted for coarse aggregate and vice versa. This was because the coarse aggregate was mainly responsible for the strength of the concrete mix. It was also noted that the enhancement in the strength of the concrete specimen was due to the rougher surface of the aggregate, which provided better bonding with the cement particles. Another reason was improper compaction of a large amount of EPS contents. As a result, the number of voids in concrete increased. So, the substitution of coarse aggregate with 15% EPS and 2% steel fibers had a minimal effect on the concrete performance. Furthermore, adding cementitious materials such as silica fume can increase the viscosity of the cement paste, making it less prone to segregation. As a result, strength increases. This also helps in maintaining a uniform distribution of aggregate throughout the mixture.

3.2.2. Compressive Strength vs. Density

Table 7 presents the compressive strength of normal concrete in comparison with various proportions of EPS and steel fibers corresponding to density at 28 days. With the addition of 15% EPS and 0% steel fibers, the reduction in density was about 20% and the compressive strength was 28% in comparison to normal concrete, as shown in Figure 7. This could be due to the high hydrophobic nature of EPS, which diminished the bonding between the cement paste and EPS particles in the concrete mix due to segregation. Based on the results, a considerable reduction was noted in the dry density and compressive strength of EPS-based concrete compared to the normal mix. However, it was also apparent that utilizing 15% EPS and 2% steel fibers improved the strength with lower density in comparison to the remaining proportions corresponding to normal concrete. The results showed that at an optimal level of 15% EPS and 2% steel fibers, concrete exhibited 10% less compressive strength with 17% lower density compared to the normal concrete mix.

This gain in compressive strength was likely caused by the inclusion of steel fibers, which increased the resistance to segregation and held the ingredients of concrete. Therefore, despite the slight reduction in strength, adopting 15% EPS and 2% steel fibers could partially compensate for the acceptable low density.

**Table 7.** Density and compressive strength of normal concrete compared with various contents of EPS and steel fibers.

| Sample No. | Concrete with EPS (%) | Steel Fibers (%) | Density (kg/m³) | Compressive Strength (MPa) |
|---|---|---|---|---|
| 1 | 0% | 0% | 1935 | 17.240 |
| 2 | 15% | 0% | 1555 | 12.379 |
| 3 | 15% | 2% | 1615 | 15.67 |
| 4 | 15% | 4% | 1645 | 14.259 |
| 5 | 30% | 0% | 1275 | 7.393 |
| 6 | 30% | 2% | 1555 | 8.499 |
| 7 | 30% | 4% | 1665 | 8.429 |
| 8 | 45% | 0% | 925 | 5.260 |
| 9 | 45% | 2% | 1350 | 6.089 |
| 10 | 45% | 4% | 1405 | 4.629 |

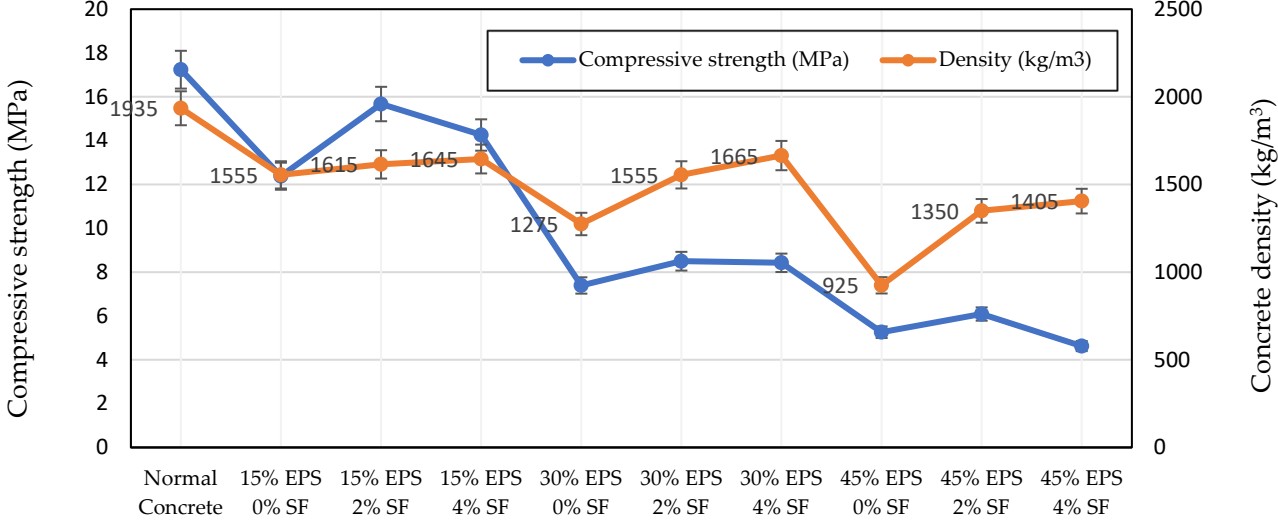

**Figure 7.** Relations between concrete density and 28 days compressive strength of EPS-based concrete.

3.2.3. Flexural Strength Test

The flexural strength tests were performed on concrete beam specimens after 28 days of curing and the results are demonstrated in Table 8. The test setup and specimen mode of failure with and without steel fibers are shown in Figure 8a,b. The flexural strength of concrete specimens featuring sole incorporation of EPS at replacement rates of 15%, 30%, and 45% for coarse aggregate exhibited values of 5.30 MPa, 5.12 MPa, and 4.47 MPa, respectively, in contrast to 6.32 MPa observed in standard concrete. Adding lightweight EPS along with steel fibers led to rising fracture strength, as indicated in Figure 9. The bend or flexural strength for all specimens varied with the different proportions of EPS and steel fibers. For instance, at the age of 28 days, the flexural strength for normal concrete was about 6.32 MPa, whereas 6.57 MPa strength was noted with the incorporation of 15% EPS and 2% steel fibers. So, the performance of lightweight EPS-based concrete with the said proportions increased by 5% compared to normal concrete. These positive bond strength effects in the lightweight EPS-based concrete were due to the ductile nature of steel

fibers, which was responsible for resistance against slipping and reduced crack initiation, increasing the bond strength of the EPS-based concrete mix as a result.

**Table 8.** The flexural strength of concrete beams with various proportions of EPS and steel fibers at 28 days.

| | % of EPS and Steel Fibers | | Flexural Strength (MPa) | | | |
|---|---|---|---|---|---|---|
| Sample No. | EPS | SF | Test 1 | Test 2 | Test 3 | Mean |
| 1 | 0% | 0% | 6.482 | 6.197 | 6.288 | 6.323 |
| 2 | 15% | 0% | 5.262 | 5.414 | 5.234 | 5.303 |
| 3 | 15% | 2% | 6.728 | 6.550 | 6.432 | 6.570 |
| 4 | 15% | 4% | 6.556 | 6.460 | 6.446 | 6.487 |
| 5 | 30% | 0% | 5.091 | 5.217 | 5.051 | 5.120 |
| 6 | 30% | 2% | 5.188 | 5.535 | 4.999 | 5.241 |
| 7 | 30% | 4% | 4.317 | 4.207 | 4.158 | 4.227 |
| 8 | 45% | 0% | 4.473 | 4.610 | 4.335 | 4.473 |
| 9 | 45% | 2% | 4.764 | 4.978 | 4.767 | 4.836 |
| 10 | 45% | 4% | 3.751 | 3.476 | 3.406 | 3.544 |

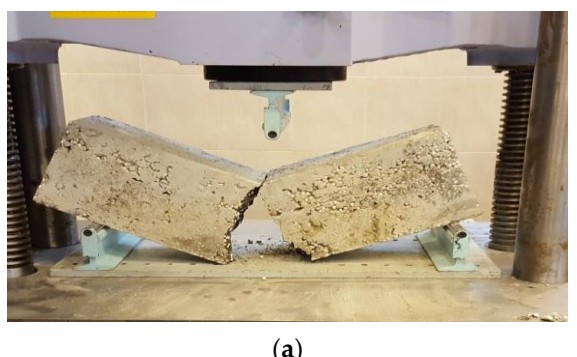

**(a)**

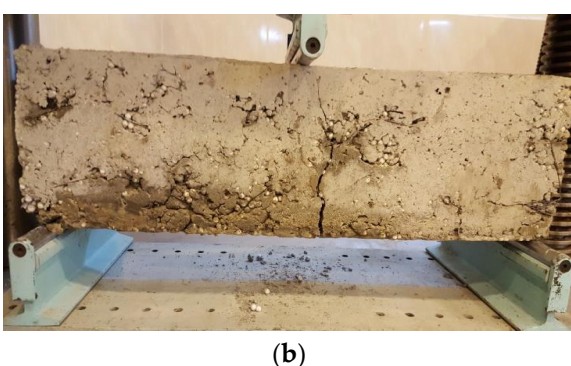

**(b)**

**Figure 8.** Flexural strength test of casted beams. (**a**) Typical EPS-based concrete beam failure; (**b**) EPS-based concrete beam failure mode reinforced with steel fibers.

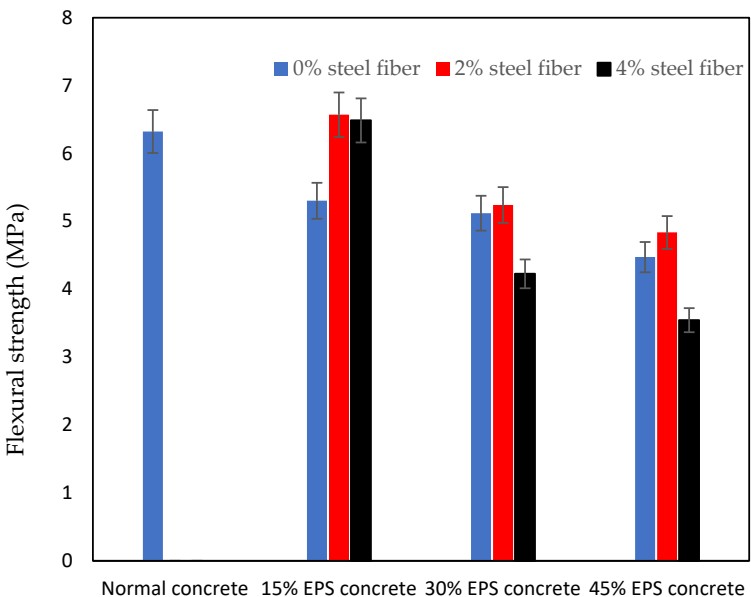

**Figure 9.** Flexural strength at 28 days of concrete with different contents of EPS and steel fibers.

It is notable that with the same content of 15% EPS and 4% steel fibers, the flexural strength improved to 3.5% compared to normal concrete. Similar results were reported: even after flexural cracking, the steel fibers could endure the elongation and expansion of cracks [33]. This is a virtue of steel fibers, which act as a bridge prior to cracking and reduce the disintegration of aggregates in concrete mixes. It could also be due to the fact that steel fibers hold crack expansion and elongation even after failure due to bending since they are made of steel. Further substitution of EPS in concrete with steel fibers showed a decrease in flexural strength. For example, with 30% and 45% incorporation of EPS and 2% steel fibers, the flexural strength was reduced to 18% and 25% in contrast to normal concrete. To conclude, the inclusion of 15% EPS in concrete with 2–4% steel fibers influenced and increased the flexural strength of concrete. Ultimately, using steel fibers in EPS-based concrete at this optimal level improved the durability and material ductility.

### 3.2.4. Flexural Strength Test vs. Density

The flexural strengths of the beam specimens after 28 days against density with different contents of EPS and steel fibers are shown in Table 9. The stress at failure in bending of concrete was established and reduced linearly with the addition of only lightweight EPS contents corresponding to density. However, when 15% EPS and 2% steel fibers were introduced, a 5% enhancement in the flexural strength of EPS-based concrete was observed alongside a remarkable 17% reduction in density compared to standard concrete. Further, at 15% EPS and steel fiber content up to 4%, there was a 3.5% improvement in strength coupled with a 15% decrease in density compared to the normal concrete benchmark, as evidenced in Figure 10.

**Table 9.** Density and flexural strength of normal concrete compared with various contents of EPS and steel fibers.

| Sample No. | Concrete with EPS (%) | Steel Fibers (%) | Density (kg/m$^3$) | Flexural Strength (MPa) |
|---|---|---|---|---|
| 1 | 0% | 0% | 1935 | 6.323 |
| 2 | 15% | 0% | 1555 | 5.303 |
| 3 | 15% | 2% | 1615 | 6.570 |
| 4 | 15% | 4% | 1645 | 6.487 |
| 5 | 30% | 0% | 1275 | 5.120 |
| 6 | 30% | 2% | 1555 | 5.241 |
| 7 | 30% | 4% | 1665 | 4.227 |
| 8 | 45% | 0% | 925 | 4.473 |
| 9 | 45% | 2% | 1350 | 4.836 |
| 10 | 45% | 4% | 1405 | 3.544 |

The increment in the flexural strength could be due to ductility in the tension of steel fibers in the mix, and with the lightweight nature of EPS, the density was reduced. The results also revealed that the concrete beam specimens with the highest steel fiber content reduced the crack propagation since the fibers are made of steel. Additionally, beyond 15% EPS and 4% steel fibers, there was a continuous reduction in the strength and density of specimens. This could be due to substituting a large content of coarse aggregate with EPS. Similarly, with large contents of steel fibers, challenges arise in terms of its compaction and overall flexural strength can be affected as a result. In summary, the inclusion of up to 15% EPS and 2% to 4% steel fibers yielded favorable outcomes, as shown in the results, with 3.5% to 5% increases in flexural strength and a simultaneous density reduction ranging between 15% to 17% relative to the mechanical characteristics exhibited by standard concrete.

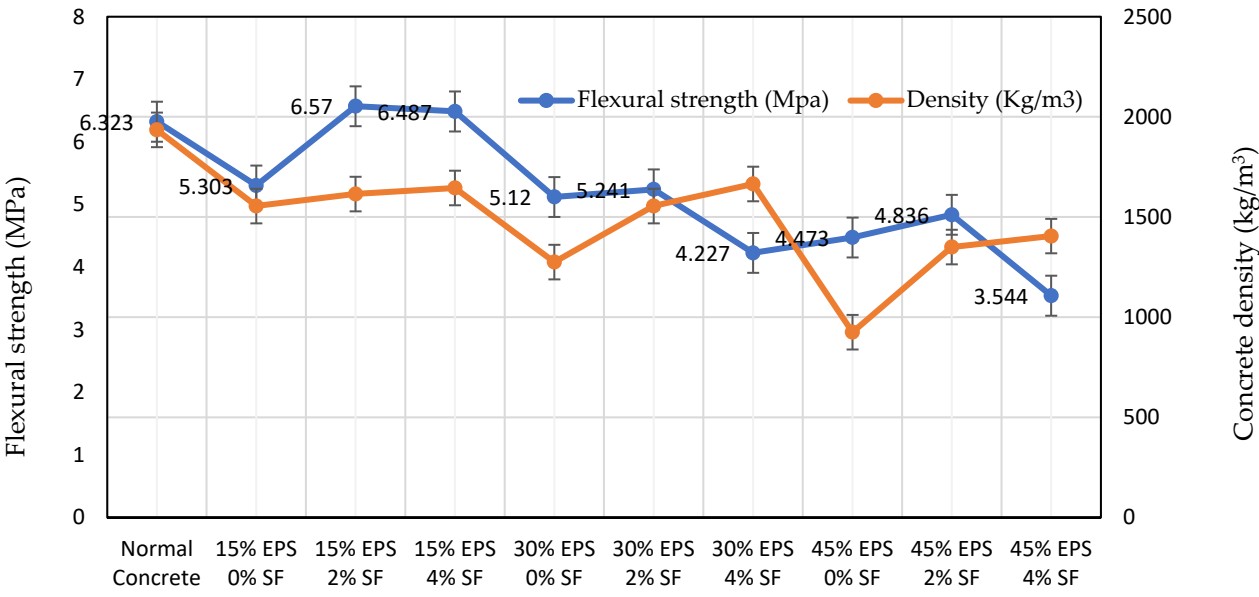

Proportions of EPS & steel fibers

**Figure 10.** Relations between hardened concrete density and flexural strength of EPS-based concrete.

3.2.5. Specific Strength of Concrete Specimens

The concrete strength-to-weight ratio is a comparison of its strength in relation to how much it weighs. In this study, to obtain the specific strength of concrete specimens, the compressive and flexural strengths of concrete with different EPS and steel fiber contents were divided by its density, as illustrated in Table 10.

**Table 10.** Strength-to-weight ratio comparison of concrete with EPS and steel fiber contents.

| Concrete with EPS (%) | Steel Fibers (%) | Saturated Dry Density (kg/m³) | 28 Days Compressive Strength (MPa) | Specific Strength (kPa-m³/kg) | 28 Days Flexural Strength (MPa) | Specific Strength (kPa-m³/kg) |
|---|---|---|---|---|---|---|
| 0% | 0% | 1935 | 17.240 | 8.909 | 6.323 | 3.267 |
| 15% | 0% | 1555 | 12.379 | 7.960 | 5.303 | 3.410 |
| 15% | 2% | 1615 | 15.67 | 9.702 | 6.570 | 4.068 |
| 15% | 4% | 1645 | 14.259 | 8.668 | 6.487 | 3.943 |
| 30% | 0% | 1275 | 7.393 | 5.798 | 5.120 | 4.015 |
| 30% | 2% | 1555 | 8.499 | 5.465 | 5.241 | 3.370 |
| 30% | 4% | 1665 | 8.429 | 5.062 | 4.227 | 2.538 |
| 45% | 0% | 925 | 5.260 | 5.686 | 4.473 | 4.835 |
| 45% | 2% | 1350 | 6.089 | 4.510 | 4.836 | 3.582 |
| 45% | 4% | 1405 | 4.629 | 3.294 | 3.544 | 2.522 |

**4. Conclusions**

The study aimed to create lightweight concrete using varied proportions of expanded polystyrene (EPS), steel fibers, and silica fume. Incorporating 15% EPS without steel fibers resulted in a 20% reduced slump compared to standard concrete due to EPS segregation. The addition of steel fibers affected workability. Density decreased by 20%, 35%, and 53%, with 15%, 30%, and 45% EPS inclusion, respectively.

The optimal balance between low density and compressive strength in EPS-based concrete was achieved with 15% EPS and 2% steel fibers. This mix showed a slight 10% decrease in compressive strength and a 17% reduction in density compared to standard concrete. Importantly, steel fibers improved resistance to segregation in the concrete mixes. Steel fibers at 2% to 4% alongside 15% EPS improved flexural strength by around

5% compared to regular concrete due to their ductile properties slowing crack growth. However, beyond 4% steel fibers, a diminishing trend was noticed. Silica fume notably enhanced concrete strength by refining the microstructure near the aggregate surface. Thus, it can be concluded that the optimal mix design for achieving the desired mechanical properties is incorporating 15% lightweight expanded polystyrene and 2% to 4% steel fibers.

However, there is scope for additional research related to integrating EPS materials in concrete across various proportions. Further studies could be conducted to explore the split tensile strength, environmental impact, and chemical interaction between EPS and steel fibers within the concrete matrix on an optimized mix ratio. This will help advance the development of lightweight concrete with enhanced performance and durability.

**Author Contributions:** Conceptualization, S.J.S., A.N. and F.H.; methodology, S.J.S., A.N., F.H. and W.A.M.; software, S.J.S., A.N. and W.A.M.; validation, S.J.S., A.N. and F.H.; formal analysis, S.J.S. and A.H.; investigation, S.J.S. and A.H.; resources, S.J.S., A.N., F.H., W.A.M. and A.H.; data curation, S.J.S., A.N. and A.H.; writing—original draft preparation, S.J.S.; writing—review and editing, S.J.S., A.N., W.A.M. and F.H.; visualization, S.J.S.; supervision, F.H.; project administration, S.J.S. and F.H.; funding acquisition, S.J.S., W.A.M. and F.H. All authors have read and agreed to the published version of the manuscript.

**Funding:** The authors acknowledge the Higher Education Commission (HEC) of Pakistan for providing the necessary funds and resources for the PhD studies of the first author under the HRDI-UESTPs/UETs Scholarship, Phase-I, Batch-VI with ref. no. 4893/2019.

**Data Availability Statement:** The data presented in this article can be obtained from the first author upon request.

**Acknowledgments:** The authors acknowledge the Department of Civil Engineering, Balochistan University of Information Technology, Engineering and Management Sciences (BUITEMS), Quetta, for use of the laboratory equipment in this research and for valuable support during the experiments and data analysis.

**Conflicts of Interest:** The authors declare no conflicts of interest. The funders had no role in the study's design; in the collection, in the analysis, or interpretation of data; in the writing of the manuscript; or in the decision to publish the results.

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
