# Peer review of "Experimental Investigation of Mechanical Properties of Concrete Mix with Lightweight Expanded Polystyrene and Steel Fibers"

_2673-4109, doi:10.3390/civileng5010011_

Round 1

Reviewer 1 Report

Comments and Suggestions for Authors

The following comments can be done to improve the manuscript.

Abstract

·         The abstract did not give clear idea about the result of the manuscript.

·         Line 32-35 rewrite the statement is not clear.

Introduction

·         Line 44 “ The strength and  bulk density of LWC ranges from 7 to 18 MPa   “ based on what this statement is not true

·         Page 3  line 106-107 is not clear rewrite/

·         Page 3 line 120-122 delete

·         Page 9 line 250-251 rewrite

·         Page 9 line 258 “On the contrary, it observed that normal specimens had fewer voids and more compact due to proper 257 integration of aggregates cement bond as shown in Fig. 5” this Fig. does not show the voids you should have a picture to the inner structure of the sample.

·         Page 9 line 269-271 rewrite/

·         Page 10 line 276-279 the statement is not clear

·         Page 11 line291-293 rewrite

·         Page 11 line 287, 295 units psi, MPa

·         Page 11 line 308-310 “Furthermore, the presence of ultra-fine material like silica fume will better fill voids between cement particles and results in a very dense concrete with higher compressive strength and extremely low permeability” this is irrelevant for the flexural section

Conclusion

By analyzing, the EPS concrete failure, it has been observed that concrete fails in two stages, at the first phase the concrete undergoes micro cracking, which is then followed by macro crack propagation. This failure pattern is unlike for all the percentage of the polystyrene particles used. This in term relates that the proportioning of the expanded polystyrene has a significant impact on the overall flexural strength of the lightweight concrete failure”

I did not see any section that analyzed the cracking behavior or any figures that support this statement.

Another comments

-error bars must be added to the figures.

-More discussion in depth must be made with the related literature

Comments on the Quality of English Language

the English language need improvement 

Author Response

#

Comment/revision

1.

Note: All page numbers are according to revised manuscript.

Abstract

The abstract did not give clear idea about the result of the manuscript.

Line 32-35 rewrite the statement is not clear.

We appreciate the reviewer's feedback on the abstract. Acknowledging the lack of clarity in the initial version, we've diligently revised the abstract to provide a clearer understanding of the manuscript's results and findings. Specifically addressing the concern about lines 37-39 according to revised manuscript. The revised abstract has been highlighted in the revised manuscript.

The demand for lightweight aggregates in the composition of concrete for diverse structural and non-structural applications in contemporary building constructions has increased. This is to achieve a controllable low density of lightweight concrete, resulting in a reduction of the overall structural weight. However, the challenge lies in achieving an appropriate strength in lightweight concrete while maintaining a lower unit weight. This research aims to evaluate the performance of lightweight concrete by integrating Expanded Polystyrene (EPS) as a partial replacement for coarse aggregate. Test specimens were cast by blending EPS with coarse aggregates at varying proportions of 0%, 15%, 30%, and 45%, while maintaining a constant water-to-binder ratio of 0.60. To enhance the bonding and structural capabilities of the proposed lightweight concrete mixes, reinforcement with 2% and 4% steel fibers by volume of the total concrete mix was incorporated. Silica fumes were introduced into the mix to counteract the water hydrophobicity of EPS material and enhance the durability of lightweight concrete, added at a rate of 10% by weight of cement in all specimens. A total of 60 samples, including cylinders and beams, were prepared and cured over a 28-day period. The physical and mechanical properties of the lightweight EPS-based concrete were systematically examined through experimental testing and compared against a standard concrete mix. The analysis of the results indicates that EPS concrete exhibits a controllable low density. It also reveals that the incorporation of reinforced material such as steel fiber enhances the overall strength of lightweight concrete.

2.

Introduction

Line 46-48 “The strength and bulk density of LWC ranges from 7 to 18 MPa “based on what this statement is not true.

Thank you for mentioning the issue related to the line 46-48 according to the revised manuscript. It has been rephrased for improved clarity according to the reference provided and highlighted in the revised manuscript as well.

According to a study, the compressive strength and bulk density of low-strength but lightweight concrete ranges from 7 to 18 MPa and 800 to 1400 kg/m3, respectively [1].

3.

Page 3 line 105-115 is not clear rewrite.

Thank you for mentioning the issue related to lines 105-115. It has been revised for improved clarity. It is highlighted in the revised manuscript.

This study explores lightweight EPS as a substitute for coarse aggregates in low-density, low-strength concrete. However, freshly mixed EPS-based concrete faces challenges like segregation due to its lightweight and hydrophobic nature, impacting workability. Optimal aggregate replacement levels affect concrete strength, and excessive substitution weakens it. Investigating EPS as a substitute is promising, but ongoing research mainly focuses on reducing density, posing strength preservation challenges. Our study assesses the structural performance of lightweight concrete properties with varying EPS levels (0%, 15%, 30%, 45%), steel fibres, and silica fume, aiming to address segregation and enhance bonding. The findings highlight that EPS concrete with steel fibres and silica fumes shows potential with low density and higher flexural strength, and an optimal lightweight concrete design mix is proposed.

Page 3 line 114-115 delete.

Thank you for mentioning the issue related to the said line. It has been deleted in the revised manuscript.

Page 9 line 239-242 rewrite.

Thank you for mentioning the issue related to the line 239-242, it has been revised for improved clarity. It highlighted in the revised manuscript.

The incorporation of EPS solely resulted in a reduction of compressive strength within the test specimens. It decreased by 32%, 50%, and 71.1% in sample number 2 , 5 and 8, respectively, comparison to standard concrete, as illustrated in Figure 6.

4.

Page 9 line 258 “On the contrary, it observed that normal specimens had fewer voids and more compact due to proper integration of aggregates cement bond as shown in Fig. 5” this Fig. does not show the voids you should have a picture to the inner structure of the sample.

Thank you for mentioning the issue related to the line 258. The author didn’t study the part of the inner structure of the specimens. It is not in the scope. So, the sentence has been deleted in the revised manuscript.

Reviewer 2 Report

Comments and Suggestions for Authors

This study presents the controllable low density of EPS concrete and the significant effect of steel fibers on the flexural strength of EPS concrete. The primary goal is to assess the physical and mechanical properties of concrete specimens containing different levels of EPS (0% to 45%) incorporated with coarse aggregates and steel fibers. The structure and content of the article are detailed and the experimental design is reasonable. However, for the article to be suitable for publication, the following revisions are suggested:

a)         The authors should emphasize the value and significance of this study in the final part of the abstract section.

b)        The Results and Discussion section should include an optimization of the mix ratio based on the analysis of the experimental results.

c)         In the materials section, the basis for choosing EPS as 2% and 4% should be succinctly listed in lines 43-44.

d)        Consider discussing the splitting tensile strength as an important indicator to characterize the strength of concrete in this article.

e)      The results section should include a discussion of the age as an important indicator of concrete performance.

Author Response

Response to Reviewer #2

#

Comment/revision

1.

The authors should emphasize the value and significance of this study in the final part of the abstract section.

We agree with the reviewer on this statement. However, we've diligently revised the abstract to better understand the manuscript's significance.

 The demand for lightweight aggregates in the composition of concrete for diverse structural and non-structural applications in contemporary building constructions has increased. This is to achieve a controllable low density of lightweight concrete, resulting in a reduction of the overall structural weight. However, the challenge lies in achieving an appropriate strength in lightweight concrete while maintaining a lower unit weight. This research is aimed at evaluating the performance of lightweight concrete by integrating Expanded Polystyrene (EPS) as a partial replacement for coarse aggregate. Test specimens were cast by blending EPS with coarse aggregates at varying proportions of 0%, 15%, 30%, and 45%, while maintaining a constant water-to-binder ratio of 0.60. To enhance the bonding and structural capabilities of the proposed lightweight concrete mixes, reinforcement with 2% and 4% steel fibers by volume of the total concrete mix was incorporated. Silica fumes were introduced into the mix to counteract the water hydrophobicity of EPS material and enhance the durability of lightweight concrete, added at a rate of 10% by weight of cement in all specimens. A total of 60 samples, including cylinders and beams, were prepared and cured over a 28-day period. The physical and mechanical properties of the lightweight EPS-based concrete were systematically examined through experimental testing and compared against a standard concrete mix. The analysis of the results indicates that EPS concrete exhibits a controllable low density. It also reveals that the incorporation of reinforced material such as steel fiber enhances the overall strength of lightweight concrete.

2.

The Results and Discussion section should include an optimization of the mix ratio based on the analysis of the experimental results.

We would like to thank the reviewer for this comment and suggestion. This test was out of the scope of this study. However, it has been revised and highlighted in the recommendation part for future research in lines 375-380 in the revised manuscript.

However, there is scope for additional research related to integrating EPS materials in concrete across various proportions. Further studies could be conducted to explore the split tensile strength, environmental impact and chemical interaction between EPS and steel fibres within the concrete matrix on an optimised mix ratio. This will help advance the development of lightweight concrete characterized by enhanced performance and durability. 

3.

In the materials section, the basis for choosing EPS as 2% and 4% should be succinctly listed in lines 130-132.

We would like to thank the reviewer for this comment and suggestion. This has been shifted as suggested by the reviewer and highlighted in the revised manuscript in lines 130-132.

In addition, to reduce the brittleness of the concrete mixture, steel fibers (SFs) of 1 mm in diameter and 50.8 mm in length were added with 2% and 4% by weight of mixed.

4.

Consider discussing the splitting tensile strength as an important indicator to characterize the strength of concrete in this article.

We would like to thank the reviewer for this comment; this test was out of the scope of this study; however, it’s been mentioned and highlighted in the recommendation for future research in lines 376-380.

Further studies could be conducted to explore the split tensile strength, environmental impact and chemical interaction between EPS and steel fibres within the concrete matrix on an optimised mix ratio. This will help advance the development of lightweight concrete characterized by enhanced performance and durability. 

5.

The results section should include a discussion of the age as an important indicator of concrete performance.

The reviewer's comment is valid. However, due to the limitation of the equipment, the research was only able to investigate the strength gain for a maximum curing period of 28 days.

Reviewer 3 Report

Comments and Suggestions for Authors

The authors present an investigation on the mechanical properties of a concrete mix with EPS and steel fibers. Concrete with different percentage of the added materials were produced and tested for compression and bending. Density and workability were also assessed. The results for the different materials were compared. Overall, the article is relatively well written, and the structure is good, but there are some grammatical and orthographic errors that must be corrected. Furthermore, the order of the tables and figures must be in the order they are mentioned in the text for better readability. Next, Table 3 needs some gridlines so that it is clear which EPS % is correct. It is also somewhat unusual not to use SI units. Since the investigation focuses on lightweight materials, it could be interesting to see the compressive and flexural strength in terms of density. Will the proposed material perform better in terms of strength pr density? If the authors address these issues, the manuscript can be published.

Comments on the Quality of English Language

As mentioned, the language is overall good, but some grammatical and orthgraphic errors must be corrected. 

Author Response

Response to reviewer #3

#

Comment/revision

1.

The authors present an investigation on the mechanical properties of a concrete mix with EPS and steel fibers. Concrete with different percentages of the added materials were produced and tested for compression and bending. Density and workability were also assessed. The results for the different materials were compared. Overall, the article is relatively well written, and the structure is good, but there are some grammatical and orthographic errors that must be corrected. Furthermore, the order of the tables and figures must be in the order they are mentioned in the text for better readability.

Next, Table 3 needs some gridlines so that it is clear which EPS % is correct.

We would like to thank the reviewer for this suggestion. We agree with the reviewer that the table values did not read well. This has been changed and modified as suggested by the reviewer and highlighted in the revised manuscript.

2.

It is also somewhat unusual not to use SI units.

We would like to thank the reviewer for mentioning the issue related to using the SI units. All the units changed to SI units in the revised manuscript. 

3.

Since the investigation focuses on lightweight materials, it could be interesting to see the compressive and flexural strength in terms of density. Will the proposed material perform better in terms of strength pr density? If the authors address these issues, the manuscript can be published. As mentioned, the language is overall good, but some grammatical and orthographic errors must be corrected.

We would like to thank you the reviewer for this suggestion. The tables and graphs b/w the compressive strength and flexural strength in terms of density has been added and highlighted in the revised manuscript.

First, the reviewer can find the graphs b/w the compressive strength vs density in the result and discussion part in line 275-293.

Compressive strength vs Density

Table 7 represent the compressive strength of normal concrete in comparison with different content of EPS & steel fiber and the corresponding density at 28 days. The addition of 15% EPS and 0% steel fibers, the reduction in density was about 20% and compressive strength of 28% in comparison to the normal concrete as shown in Fig. 7. It could be due to highly hydrophobic nature of EPS which weaken the bonding b/w the cement paste and EPS particles in concrete mix due to segregation. Based on the results, there is a substantial decrease in the dry density and compressive strength compared to normal mix. However, it is also apparent that utilizing 15% EPS and 2% steel fiber improved the strength with lower density in comparison to remaining of all proportions corresponding normal concrete. It shows that at optimal level of 15% EPS and 2% steel fibers concrete exhibited a 10% lesser compressive strength with 17% low density corresponding to normal concrete mix. This gain in strength is likely due to the inclusion of steel fibers which increased the resistance to segregation and hold the ingredients of concrete. Therefore, despite the slight reduction in the strength, adopting 15% EPS and 2% steel fibers could partially compensate with acceptable low density.       

Table 7

Density and compressive strength of normal concrete compared with different content of EPS & steel fiber

Sample

No.

Concrete with EPS (%)

Steel fibers (%)

Density (kg/m3)

Compressive strength (MPa)

1

0%

0%

1935

17.240

2

15%

0%

1555

12.379

3

15%

2%

1615

15.67

4

15%

4%

1645

14.259

5

30%

0%

1275

7.393

6

30%

2%

1555

8.499

7

30%

4%

1665

8.429

8

45%

0%

925

5.260

9

45%

2%

1350

6.089

10

45%

4%

1405

4.629

Fig. 7. Relations b/w hardened concrete density and 28 days compressive strength of EPS concrete

Second, the reviewer can find the graphs b/w the Flexural strength vs density in the result and discussion part in line 329-354.

Flexural strength vs Density

The flexural strength of beam specimen after 28 days of curing against density with different contents of EPS and steel fibers are shown in Table 9. The flexural strength of concrete is found to decrease linearly with the addition of only lightweight EPS contents in corresponding to density. However, when 15% EPS and 2% steel fibers are introduced, a 5% enhancement in the flexural strength of EPS concrete is observed alongside a remarkable 17% reduction in density compared to standard concrete. Further, at 15% EPS and steel fiber content up to 4%, there is a 3.5% improvement in strength coupled with a 15% decrease in density compared to the normal concrete benchmark, as evidenced in Figure 10.

The increment in the flexural strength could be due to ductility in tension of steel fibers in the mix and with lightweight nature of EPS the density reduced. It was also found that adding steel fibers could reduce crack initiation and increase the strength of EPS concrete since it made of steel. Additionally, byond 15%EPS and 4% steel fibers, there in continuous reduction in the strength and density of specimens. It could be due to substitution of large content of coarse aggregates to EPS, which is mainly responsible for strength in concrete mix. Similarly, with the large contents of steel fibers, it should also be taken into account that it can reduce the compactness of the concrete matrix and negatively impact on flexural strength. In summary, the inclusion of up to 15% EPS and 2% to 4% steel fibers yielded favorable outcomes, as shown in results that 3.5% to 5% increase in flexural strength and simultaneously density reduction ranging between 15% to 17% relative to the mechanical characteristics exhibited by standard concrete.

Table 9

Density and flexural strength of normal concrete compared with different content of EPS & steel fiber

Sample

No.

Concrete with EPS (%)

Steel fibers (%)

Density (kg/m3)

Flexural strength (MPa)

1

0%

0%

1935

6.323

2

15%

0%

1555

5.303

3

15%

2%

1615

6.570

4

15%

4%

1645

6.487

5

30%

0%

1275

5.120

6

30%

2%

1555

5.241

7

30%

4%

1665

4.227

8

45%

0%

925

4.473

9

45%

2%

1350

4.836

10

45%

4%

1405

3.544

Fig. 10. Relations b/w hardened concrete density and flexural strength of EPS concrete.

Reviewer 4 Report

Comments and Suggestions for Authors

Dear authors,

Here are some major questions that should be take in consideration related to this topic:

  1.  

    • - How does the inclusion of lightweight expanded polystyrene affect the compressive strength of the concrete?
    • - What is the influence of steel fibers on the compressive strength of the concrete mix?
  2.  

    • - What is the flexural strength of the concrete mix with lightweight expanded polystyrene, and how does it compare to traditional concrete mixes?
    • - How do steel fibers contribute to the flexural performance of the concrete?
  3.  
    •  
  4.  

    • - What impact does the combination of lightweight expanded polystyrene and steel fibers have on the durability of the concrete, especially in terms of resistance to freeze-thaw cycles, chemical exposure, and other environmental factors?
  5.  

    • - What is the bond strength between the lightweight expanded polystyrene particles, steel fibers, and the concrete matrix?
    • - How does the bond strength influence the overall performance of the concrete?
  6.  
  7.  

    • - What is the optimal mix design for achieving the desired mechanical properties while incorporating lightweight expanded polystyrene and steel fibers?
    • - How do different proportions of materials affect the performance of the concrete?
    • Best regards!

Author Response

Response to reviewer #4

#

Comment/revision

1.

• How does the inclusion of lightweight expanded polystyrene affect the compressive strength of the concrete?

We would like to thank the reviewer for this comment. The compressive strength diminished with the addition of only the presence of EPS in test specimens. It could be due to the lightweight and hydrophobic nature of EPS, which tends to reduce the water absorption capacity of the concrete ingredients, and EPS particles floated up as a result of segregation and bleeding occurred.

•What is the influence of steel fibers on the compressive strength of the concrete mix?

We would like to thank the reviewer for this comment. Steel fibres slightly amplified the compressive strength by holding all the ingredients of EPS-based concrete specimens and increased the resistance to segregation.

2.

• What is the flexural strength of the concrete mix with lightweight expanded polystyrene, and how does it compare to traditional concrete mixes?

We would like to thank the reviewer for this comment. The flexural strength of concrete slightly reduced with the substitution of EPS only. Furthermore, we used the standard concrete mix as a benchmark for comparison of flexural strength for newly proposed EPS-based concrete with different proportions.

• How does steel fibers contribute to the flexural performance of the concrete?

We would like to thank the reviewer for this comment. The flexural strength of EPS-based concrete increases with the inclusion of steel fibres. This is because the bridging effect of the SFs before the cracking reduced the fracture of the coarse, lightweight aggregate. It could be due to the fact that the steel fibres are able to endure the expansion and elongation of cracks even after flexural cracking has occurred since it is made of steel.

3.

What impact does the combination of lightweight expanded polystyrene and steel fibers have on the durability of the concrete, especially in terms of resistance to freeze-thaw cycles, chemical exposure, and other environmental factors?

We would like to thank the reviewer for this valid comment. However, this test was out of scope of this study. It has been revised and highlighted in the recommendation part for future research in line 375-380 in the revised manuscript.

4.

What is the bond strength between the lightweight expanded polystyrene particles, steel fibers, and the concrete matrix?

We would like to thank the reviewer for this comment. Adding up to 15% expanded polystyrene particles and 2%-4% steel fibers in the concrete matrix had a positive impact on bonding, resulting in a 3-5% increase in flexural strength and reduced the density from 15%-17% compared to the standard concrete mix.

5.

How does the bond strength influence the overall performance of the concrete?

We would like to thank the reviewer for this comment. The positive bond strength effects in the lightweight EPS concrete are due to the ductile nature of steel, which was responsible for resistance against slipping and reduced crack initiation. As a result, it increased the bond strength of EPS-based concrete.

6.

What is the optimal mix design for achieving the desired mechanical properties while incorporating lightweight expanded polystyrene and steel fibers?

We would like to thank the reviewer for this comment. The optimal mix design for achieving the desired mechanical properties incorporates 15% lightweight expanded polystyrene and 2% to 4% steel fibers. In addition, the utilization of silica fume can contribute to strength development at appropriate content.

7.

How do different proportions of materials affect the performance of the concrete?

We would like to thank the reviewer for this comment. The presence of 2-4% steel fibers with 15% lightweight EPS enhances the mechanical property of EPS-based concrete corresponding to the standard concrete mix. The strength reduced as the proportions of EPS increased from 15% to 30% and 45%. 

Round 2

Reviewer 1 Report

Comments and Suggestions for Authors

Accept after revision 

Author Response

Dear Reviewer,

Thank you very much.

Your review and comments were really helpful for the improvement of our paper.

Reviewer 2 Report

Comments and Suggestions for Authors

The submission has been greatly improved and is worthy of publication.

Author Response

(The authors gave the same response as above.)

Reviewer 3 Report

Comments and Suggestions for Authors

The authors have taken all the reviewers’ comments into account in the revised manuscript. It was especially interesting to see the density variation. A typical value to present would be the compressive and flexural strength divided by the density. Then it would be easier to see which mix performs better in terms of strength. It is up to the authors if they want to add this. Furthermore, the revision has caused a division over two pages of Table 4, Table 5, Table 6, and Table 8. I would advise to fix this before publication.

Comments on the Quality of English Language

The quality of English language is fine

Author Response

Dear Reviewer,

Thank you very much.

Your review and comments were really helpful for the improvement of our paper.

Table 10 has been added as per your comments.

Furthermore, we made the changes to all tables to appear on the same page. 

Reviewer 4 Report

Comments and Suggestions for Authors

No other comments.

Author Response

(The authors gave the same response as above.)
